# White Rot Fungi as Tools for the Bioremediation of Xenobiotics: A Review

**DOI:** 10.3390/jof10030167

**Published:** 2024-02-21

**Authors:** Giselle Torres-Farradá, Sofie Thijs, Francois Rineau, Gilda Guerra, Jaco Vangronsveld

**Affiliations:** 1Department of Microbiology and Virology, Faculty of Biology, University of Havana, Calle 25 No. 455. Vedado, Habana 10400, Cuba; ggr@fbio.uh.cu; 2Centre for Environmental Sciences, Hasselt University, Agoralaan, Building D, Diepenbeek, B-3590 Hasselt, Belgium; sofie.thijs@uhasselt.be (S.T.); francois.rineau@uhasselt.be (F.R.); jaco.vangronsveld@uhasselt.be (J.V.); 3Department of Plant Physiology and Biophysics, Institute of Biological Sciences, Marie Curie-Sklodowska University, Akademicka 19, 20-033 Lublin, Poland

**Keywords:** xenobiotics, white rot fungi, ligninolytic enzymes, bioremediation

## Abstract

Industrial development has enhanced the release into the environment of large quantities of chemical compounds with high toxicity and limited prospects of degradation. The pollution of soil and water with xenobiotic chemicals has become a major ecological issue; therefore, innovative treatment technologies need to be explored. Fungal bioremediation is a promising technology exploiting their metabolic potential to remove or lower the concentrations of xenobiotics. In particular, white rot fungi (WRF) are unique microorganisms that show high capacities to degrade a wide range of toxic xenobiotic compounds such as synthetic dyes, chlorophenols, polychlorinated biphenyls, organophosphate pesticides, explosives and polycyclic aromatic hydrocarbons (PAHs). In this review, we address the main classes of enzymes involved in the fungal degradation of organic pollutants, the main mechanisms used by fungi to degrade these chemicals and the suitability of fungal biomass or extracellular enzymes for bioremediation. We also exemplify the role of several fungi in degrading pollutants such as synthetic dyes, PAHs and emerging pollutants such as pharmaceuticals and perfluoroalkyl/polyfluoroalkyl substances (PFASs). Finally, we discuss the existing current limitations of using WRF for the bioremediation of polluted environments and future strategies to improve biodegradation processes.

## 1. Introduction

Environmental pollution with xenobiotics compounds is one of the most critical concerns on the planet; nowadays, research on pollutants, their origin and possible solutions are scientific priorities.

The word xenobiotic comes from the Greek word xenos, the meaning of which is “foreign” or “strange”; xenobiotics are organic compounds showing atypical structural characteristics and that are foreign to cellular metabolism. The existence of any compound in high concentrations can also be considered xenobiotic, for instance, the presence of pharmaceutical drugs in the human body which are not produced by the body itself or are a normal part of the diet. Moreover, these synthetic chemicals have very complex chemical structures and are very resistant to photolytic processes and biodegradation by indigenous microorganisms [1,2].

The major sources of xenobiotics are textiles (synthetic dyes), pharmaceutics (pharmaceutically active compounds (PhACs)), paper (paper and pulp effluents), plastic (polyvinyl chloride), the food (food additives, lecithin) and petroleum (benzene, xylene) industries and agriculture (pesticides) [3,4,5]. Moreover, polycyclic aromatic hydrocarbons (PAHs) are a wide and heterogeneous class of toxic organo-pollutants, and they originate from the incomplete combustion of organic matter including the burning of fossil fuels, coal tar, waste incineration and petroleum spills and discharge [6]. Once xenobiotics are released into the environment, they can enter the food chain, causing harmful impacts at each trophic level, and can adversely affect human and animal health due to their toxicity, mutagenicity, carcinogenicity and teratogenic effects in humans [7,8,9]. The presence of xenobiotic compounds have been reported from daily use products to agricultural products. Humans are exposed to these recalcitrant chemicals through inhalation, ingestion (through the consumption of contaminated water, fruits, vegetables, meat and fish) and adsorption through the skin (e.g., cosmetic products) [4,8].

Xenobiotics persist for long time (years) in the environment. For example, in aquatic environments, hydrophobic compounds are deposited as sediments, becoming hazardous upon exposure to organisms. Any exposure to the polluted sediments affects the lower trophic levels. Through biomagnification, they can also lead to serious toxic effects at higher trophic levels. Moreover, xenobiotics can cause severe health problems and long-term effects such as cardiovascular defects, kidney and liver damage, lung irritation, neurodegeneration, autoimmune disorders, adverse reproductive problems and eventually cancer as a result of the prolonged consumption of such pollutants in food or drinks [2,8,10]. In animals, xenobiotics affect their reproduction and immune functions [9].

Therefore, the development of innovative treatment technologies for the removal of xenobiotics is required.

The traditional treatment of wastewater containing xenobiotic compounds consists of chemical and/or physical methods such as photochemical oxidation, electrochemical destruction, membrane filtration, ozonation, chemical flocculation, volatilization, ion exchange and electrokinetic coagulation [11,12,13]. Furthermore, the sorption of pollutants into activated carbon, carbon nanotubes and fullerene have been used for the treatment of polluted wastewater [14]. Even though it has been demonstrated that these techniques are efficient, they have disadvantages, such as the generation of harmful byproducts, the accumulation of high volumes of residuals and high costs [15,16,17], consequently restraining their application in environmental remediation [18].

More recently, the exploitation of biological approaches for the remediation of contaminated areas has become more and more widely recognized as a cost-effective and suitable strategy. Mycoremediation is a promising technology exploiting the metabolic potential of fungi in order to eliminate or decrease xenobiotics. In particular, white rot fungi (WRF), as organisms living on wood, have evolved to degrade major wood polymers, including lignin. Due to the broad substrate range of their ligninolytic enzymes, WRF are exceptional microorganisms that show extraordinary capacities to degrade a widespread range of lignin-related compounds such as xenobiotics, including synthetic dyes, polychlorinated biphenyls, organophosphate pesticides and polycyclic aromatic hydrocarbons [5,17,18,19,20].

In recent years, various reviews focusing on the degradation of specific categories of xenobiotics by WRF have been published; some concentrate on the degradation of specific compounds, such as synthetic dyes [21,22,23], PAHs [24,25,26] or pharmaceuticals [27,28]. Some have focused on special oxidative enzymes [29,30,31,32] or the methods for the treatment of xenobiotics [33,34,35]. However, very few papers have illustrated a global overview concerning the utilization of WRF and their enzymes for the bioremediation of xenobiotics.

In this review, we describe the main characteristics of WRF, the characteristics of the enzymes associated with the fungal degradation of organic pollutants, the main mechanisms used by fungi to degrade lignin and organic pollutants and the suitability of fungal biomass or extracellular enzymes for bioremediation, taking into account the different environments of soil or water. We also exemplify the state-of-the-art strategies exerting the role of several fungi in degrading pollutants such as dyes, PAHs and emerging pollutants such as pharmaceuticals and perfluoroalkyl/polyfluoroalkyl substances (PFASs). Finally, we have discussed the existing challenges of using WRF for the bioremediation of polluted environments and future strategies to improve biodegradation processes.

## 2. White Rot Fungi and the Degradation of Lignin

WRF are filamentous fungi from the division Basidiomycota and class Basidiomycetes [36,37]. Basidiomycetes produce basidiospores (sexual spores) [38,39] and are found in soils with high amounts of organic matter, trees, decaying wood and agricultural residues. They transform and degrade lignocellulosic compounds, playing a main role in the carbon cycle on our planet [40,41]. They have adaptive abilities that allow them to grow in and tolerate unfavorable environmental conditions, acting as natural degraders of lignocellulose [42,43].

Taking into account the pathways of basidiomycetes degrading lignocellulosic material, two general categories have been recognized: white rot fungi (WRF) and brown rot fungi (BRF), whose names originate from the whitish or brown appearance of the wood after the growth of these fungi and their degradation of it, and ultimately of the nature of the wood polymers remaining after they are degraded by basidiomycetes.

Genomic studies of wood decay organisms have concentrated on the study of model fungal systems such as *Phanerochaete chrysosporium* for white rot (in which all plant cell wall components are degraded) and *Postia placenta* and *Serpula lacrymans* for brown rot fungi (BRF) (in which lignin is modified but not appreciably degraded). However, WRF are the unique organisms capable of lignin decay. Authors such as Floudas et al. [44], Riley et al. [45], Levasseur et al. [46], Mori et al. [47] and Zhang et al. [48] have informed the study of genomes from various WRF species, including *Phanerochaete chrysosporium*, *Trametes versicolor*, *Stereum hirsutum*, *Fomitiporia mediterránea*, *Punctularia strigosozonata*, *Dichomitus squalens*, *Heterobasidion annosum*, *Phlebia acerina*, *Phanerochaete sordida* and *Cerrena unicolor*. In all of them, the presence of genes coding class II peroxidases and laccases has been detected, the principal enzymes that directly attack lignin. Most of the genomic studies correlating the white vs. brown rot modes of wood decay have typically focused on the lignin-degrading ligninolytic class II peroxidases and the hydrolytic and oxidative enzymes involved in the attack of crystalline cellulose. However, it has also been demonstrated that other enzymes, such as laccases, cellobiose dehydrogenases and potentially many others, can contribute to white rot. However, Riley et al. [45] point out that this classification of WRF and BRF does not reveal the variety of mechanisms by which decay fungi achieve their nutrition. The authors proposed limiting the phrase “white rot” to fungi capable of degrading completely all polymers from the cell walls through the action of ligninolytic enzymes in combination with enzymes that degrade cellulose [45].

Lignin constitutes the non-polysaccharide portion of lignocellulosic biomass in plants providing it with mechanical strength. Lignin is a three-dimensional natural aromatic heteropolymer complex resulting from the dehydrogenate polymerization of coumaryl alcohol, coniferyl alcohol and sinapyl alcohol [49]. Due to its complex structure, lignin is difficult to degrade. The molecular mass of lignin ranges from 600 to 1000 kDa, which makes it too large to be adsorbed by fungi for intracellular attack. Furthermore, due to the presence of various covalent bonds (carbon-carbon or ether), lignin cannot be degraded according to hydrolysis mechanisms. Due to the complexity of lignin and its phenylpropanoic polymeric structure, enzymes associated with its degradation needs to be extracellular and to have wide substrate specificity [50]. WRF are unique microorganisms capable of mineralizing lignin due to their degradative system, which is non-stereoselective, non-specific and based on free radicals due to their very high oxidative potential [49,51].

WRF produce two large groups of enzymes (Figure 1), taking into account their function in the complex multi-enzymatic system [50]. One of the groups comprises the enzymes that directly attack lignin: laccases and class II heme-containing peroxidases, including lignin peroxidases (LiPs), manganese peroxidases (MnPs) and versatile peroxidases (VPL) [52]. The simultaneous production of these enzymes is not common in all WRF; this depends on the species, indicating that not all enzymes have to be present for the degradation of lignin [51,53]. For example, the absence of basidiomycete-type laccases *sensu stricto* has been observed in the genomes of species of WRF such as *P. chrysosporium*, *Phanerochaete carnosa* and *Auricularia delicata*. Instead, these fungi have genes of the related AA1 subfamily of multicopper oxidases. Moreover, in the culture filtrates of the species of *Jaapia argillacea* and *Botryobasidium botryosum,* no laccase activity has been detected [45,54].

For example, based on their ligninolytic enzyme composition and secretion, some researchers have divided WRF into four groups, namely (1) LiP-, MnP- and laccase-producing strains; (2) MnP- and laccase-producing strains; (3) LiP- and MnP-producing strains; (4) LiP- and laccase-producing strains [29].

There is a second group of enzymes that cannot attack wood by themselves but cooperate in the degradation process of lignin. These auxiliary enzymes include aryl alcohol oxidases (AA3_2 subfamily (auxiliary activity), EC 1.1.3.7), glyoxal oxidases (GLOX, EC 1.2.3.5), dye-decolorizing peroxidases (DyPs, EC 1.11.1.19), pyranose dehydrogenases (EC 1.1.99.29), methanol oxidases (EC 1.113.13), chloroperoxidases (EC 1.11.1.10) and cytochrome c peroxidases (EC 1.11.1.5). This second group also includes enzymes involved in the intracellular production of hydrogen peroxide, including superoxide dismutases (EC 1.15.1.1) and intracellular glyoxal oxidases (EC 1.2.3.15) [49,50,55,56].

Lignin is inaccessible to the ligninolytic enzymes laccases and peroxidases since they are too big to enter wood tissue. For this reason, lignin degradation occurs in the presence of these enzymes but with the help of various low-molecular-mass compounds, called mediators [49,56]. The veratryl alcohol radical, hydroxyl radicals and Mn^3+^, among others, are able to migrate within these compounds and oxidize lignin, resulting in a destabilization of bonds and the depolymerization of the macromolecule [57]. Hydrogen peroxide also contributes to the generation of radicals of oxygen that directly attack lignin. Moreover, organic acids may be secreted as a way to prevent self-degradation on the part of WRF, as they are known to chelate and stabilize Mn^3+^ [50].

The cooperation of the two groups of enzymes described above, together with compounds of low molecular mass and radicals, makes clear how the fungi can degrade lignin, a substrate normally recalcitrant to microbial attack.

Several authors have elucidated that the principal enzymes associated with the degradation of xenobiotics by WRF are lignin-modifying enzymes such as laccases, LiPs, MnPs and VPLs. However, in recent years, an association between fungal enzymes such as dye-decolorizing peroxidases and cytochrome P450 monooxygenases, which have an important role in the degradation of lignin-related compounds, has been described. In this review, we summarize the structure and function of the fungal enzymes associated with the degradation of xenobiotics.

## 3. Main Enzymes Associated with the Degradation of Xenobiotics

### 3.1. Structure, Function and Applications of Laccase

Laccases (AA 1. EC 1.10.3.2) (benzene diol: oxygen oxidoreductase) represent the biggest subgroup of blue multi-copper oxidases (MCOs). Laccases utilize the distinctive redox ability of copper ions to catalyze the oxidation of a diversity of aromatic substrates concomitantly with the four-electron reduction of molecular oxygen into water [56,57,58,59]. Laccases were first demonstrated in exudates of *Rhus vernicifera*, the Japanese lacquer tree, at the end of 19th century [60]. A few years later, they were also demonstrated in fungi [61]. Laccases have been isolated from bacteria, fungi, plants and some arthropods and insects [62,63]. In fungi, laccases have different physiological roles, such as lignin degradation and detoxification, plant pathogen/host interaction, morphogenesis, stress defense and pigment production [58,62,63].

More than 100 fungal laccases have been purified and characterized [16,64]. The majority of fungal laccases are extracellular monomeric globular proteins; however, oligomeric laccases (dimer or tetramer) are also known [58,65]. Laccase enzymes have approximately 550 amino acids, including 20 amino acids in the N-terminal acting as secretion signal peptides [56,65]. The molecular weights range from 50 to 130 KDa, and the extent of glycosylation usually varies between 10% and 25%, being in a few cases higher than 30% [63,66]. The carbohydrate moiety contributes to the stability of the laccase enzyme, and it is composed of residues of galactose, mannose and acetylglucosamine [63,67]. Typical fungal laccases have an acidic isoelectric point (pI) between 2.6 and 4.5. They are stable at a pH between 3 and 7.0 and temperatures below 40 °C [68].

Laccases from white rot species have a redox potential close to 800 mV, which allows the extraction of electrons from substrates [64]. Laccases can oxidize orthophenols, paraphenols, aminophenols, polyphenols, aliphatic and aromatic amines and lignin through the removal of a single electron to form a free radical. Laccases can also oxidize non-phenolic compounds in the presence of appropriate redox mediators such as the metabolites produced by the fungus or natural compounds such as veratryl alcohol, p coumaric acid and gallic acid, among others present in lignocellulosic materials [16,68]. Moreover, chemical artificial substrates of the enzymes 2,2-azino-bis 3-ethylbenzotiazoline-6-sulfonate (ABTS) and n-hydroxybenzotriazole (HBT) have been reported as excellent laccase mediators [69,70,71].

Analysis based on multiple sequence alignments of more than 100 laccases resulted in the identification of a set of four ungapped sequence regions, L1–L4, very important to classifying laccases within the broader class of MCOs [58,72,73]. Moreover, physical and structural chemistry methods such as nuclear magnetic resonance and X-ray have been used to study the three-dimensional molecular structure of laccases [74]. Laccase monomers are generally composed of three domains and have 4 copper atoms in their active site, 1 cysteine residue and 10 histidine residues directly involved in their binding. Cu1 is located in domain 3, whilst the trinuclear center (TNC) cluster is inserted between domains 1 and 3: both domains contribute residues to copper coordination. Due to their spectroscopic properties, the different copper centers can be identified. A type 1 copper ion is present at the T1 site, and its close coordination with cysteine gives as a result an intense absorption band around 600 nm and the characteristic blue color to this enzyme. The T2 site is electron paramagnetic resonance (EPR)-active and exhibits weak absorption in the visible region, whereas at the T3 site, there are two copper ions which are EPR-silent due to antiferromagnetic coupling, mediated by a linking ligand [74,75].

In the catalytic cycle of laccase, the T1 site extracts an electron from a phenolic substrate (oxidizing it into a phenoxyl radical); subsequently, the electrons are transferred to the trinuclear center (TNC) T2/T3 through to the His–Cys–His tripeptide, where dioxygen is reduced into water (Figure 2) [58]. Two water channels are present in the interior of the enzyme, whose function is to provide access to the trinuclear center T2/T3. The first channel allows access of the O_2_ to the type 2 copper atoms from the T3 site. There is a second water channel close to the type 2 copper atoms (T2) that allows the exit of the water molecules formed [62,76].

In WRF, laccases are coded by a family of genes which are differentially regulated. The production of several laccase isozymes has been observed in many species. Some of these genes coding isoenzymes are expressed constitutively, while the expression of others is induced by different aromatic compounds [58,77]. Several species produce a wide variety of isoenzymes, which may differ in their biochemical structure and properties. An evident example is found in the laccase isoenzymes produced by *Pleurotus ostreatus*. Eight different laccase isoforms are synthetized by *P. ostreatus*, six of which have been characterized [78,79,80,81,82,83]. The presence of such complex gene families and the diversity of isozymes is possibly due to the different physiological roles proposed for laccase during the fungal life cycle.

Different authors have described that several species of WRF synthetize laccases—for example, species of *Agaricus bisporus* [84], *Phlebia (Merulius) radiata* [85], *Trametes hirsuta* [86,87], *Pycnoporus cinnabarinus* [88], *Phanerochaete chrysosporium*, *Coriolopsis polyzona*, *Lentinus tigrinus* [66] *P. ostreatus* [79], *Pleurotus* spp. [89], *T. versicolor* [90], *Trametes maxima* [91], *Trametes ochracea*, *Trametes villosa* and *Trametes gallica* [66]. In the last years the production of laccases in different isoforms by different strains of the genus *Ganoderma* has been reported [92,93,94,95]. Interesting, the production of laccases by marine fungi like *Diaporthe phaseolorum* and *Pestalotiopsis* spp. has been described. These laccases are adapted to a high salt content, and for that reason, these enzymes are suitable for the treatment of industrial saline and alkaline effluents such as colored industrial wastewater containing pollutants such as pulp, paper, textile effluents and molasses-based distillery waste [96,97,98].

### 3.2. Structure and Function of Fungal Class II Peroxidases

Class II peroxidase enzymes are classified as the “non-animal peroxidase superfamily”. These enzymes belong to the CAZy family Auxiliary Activity Family 2 (AA_2) [46,99]. This group of fungal heme peroxidases includes the enzymes acting on lignin [lignin peroxidases (LiPs), manganese peroxidases (MnPs), versatile peroxidases (VPL)] [52,100].

This versatile group of enzymes shows an exceptionally broad substrate spectrum, catalyzing the oxidation of both organic and inorganic compounds in a non-specific manner using hydrogen peroxide (H_2_O_2_) as an oxidant [52,101,102,103]. Ligninolytic peroxidases have higher redox potentials compared to laccases, which allows the oxidation of a broad range of recalcitrant substrates [46].

Class II peroxidases share a high degree of structural homology, containing protoporphyrin IX (heme) as a prosthetic group, an N-terminal signal peptide, disulfide bonds and calcium-binding sites [103,104]. The crystal structures of these peroxidases show an overall compact and mostly helical fold for fungal class II peroxidases with heme tightly embedded between two domains, both of which contain one stabilizing Ca^2+^ ion. Two conserved His residues, the proximal and the distal, and distal side Arg residues are conserved for peroxidative catalytic function [52,103].

The basic reaction mechanism is identical in all heme peroxidases [105,106]. Hydrogen peroxide enters the prosthetic group via the principal heme access channel, acting as an electron acceptor, and subsequently is reduced into water. The substrates that are required for the catalytic cycle vary according to the type of peroxidase [52].

To date, a crystal structure of a ligninolytic peroxidase complexed with an aromatic substrate has not been described. It has been accepted that the main heme channel is too narrow for direct contact between the aromatic substrates and the heme cofactor [106]. Consequently, long-range electron transfer (LRET) from a protein radical at the surface of the enzyme, acting as the substrate oxidizer, to the heme cofactor has been suggested to explain how the oxidation of aromatic substrates, redox mediators and lignin takes place [102].

### 3.3. Lignin Peroxidases

Lignin peroxidases (LiPs) (EC.1.11.1.14) (diarylpropane: oxygen, H_2_O_2_, oxidoreductase) are monomeric glycoproteins with a molecular mass of 38 to 47 KDa and an optimum pH ranging from 3.0 to 4.7 and a theoretical pI of 3.3–4.7 [51,52,107]. LiPs were first described in 1983 in the fungus *P. chrysosporium* [14]. Afterward, multiple isozymes have been described in species of WRF such as *T. versicolor*, *P. radiata*, *Phlebia tremellosa*, *Bjerkandera* spp. and *Phanerochaete sordida* [14,103,107].

LiPs have a high redox potential of 1400 mV and the ability to directly oxidize non-phenolic aromatic lignin moieties and similar compounds such as the nonphenolic β-O-4 linkage type arylglycerol-aryl ethers [14,104,107]. In native producers such as the fungus *P. chrysosporium*, LiP is secreted with the natural phenolic substrate veratryl alcohol (VA; 3,4-dimethoxybenzyl alcohol). VA acts as a diffusible natural redox mediator promoting the oxidation of inaccessible substrates [108]. This mechanism facilitates the indirect interaction between the protein active site and an aromatic substrate such as lignin [52,109].

The crystal structure of a LiP isolated from *P. chrysosporium* has been described [110]. This LiP is a globular enzyme and contains three trypthophans (Trp) and eight methionines (Met). The high redox potential of the LiP is attributed to the Trp 171 in *P. chrysosporium*, as it enables the stabilization of the cations of the veratryl alcohol radicals [104]. The oxidation of the substrates by the LiP occurs via the LRET pathway attributed to Trp171, which is linked to the heme group. The Trp171 residue positioned at the surface of the enzyme is a unique characteristic of both LiPs and VPLs, while the latter use Trp164 [111,112,113,114]. Figure 3 shows the mechanism of oxidation by LiP. One-electron oxidation of the lignin models results in the formation of a reactive cation radical intermediate with prevalence for Cα–Cβ cleavage, allowing the formation of B-ring-derived aromatic derivatives, veratraldehyde and ring fission products. With the preferred aromatic electron donor, veratryl alcohol (3,4-dimethoxybenzyl alcohol), veratraldehyde is formed via LiP catalysis as a single product [103]. LiPs have also been shown to be capable of oxidizing phenolic aromatic compounds [115].

The role of LiPs in ligninolysis could be the further transformation of the lignin fragments which are initially released by MnPs. LiPs are not essential to attacks on lignin: several highly active WRF and litter-decaying fungi (e.g., *Ceriopsis subvermispora*, *Dichotomitus squalens*, *Panus tigrinus*, *Rigidosporus lignosus*) do not excrete these enzymes [103].

### 3.4. Manganese Peroxidases

Manganese peroxidases (MnPs) (EC 1.11.1.13; Mn(II): H_2_O_2_ oxidoreductase) were first described in *P. chrysosporium* over 30 years ago, but, unfortunately, at the beginning, less attention was paid to them in comparison with LiPs and laccases [107,116].

MnPs are the most frequently occurring class II peroxidases among basidiomycete fungi [117]. The molecular masses of MnPs range from 37 to 62.5 KDa. They have an isoelectric point between 3.2 and 4.6 and an optimum pH between 4.0 and 5.0. Phylogenetic analyses of the genes coding MnP enzymes have shown that there are two groups of MnPs: typical long MnP enzymes (group B) and short-type hybrid MnP variants (group A), the latter being evolutionary, more related to VPLs and LiPs than to the classical long MnPs [103,104].

MnPs need the presence of H_2_O_2_ and Mn^2+^ (which is in wood). Under acidic conditions, MnPs can oxidate Mn^2+^ ions. Chelated Mn^3+^ acts as a diffusible low-molecular-weight mediator that is able to attack phenolic structures including lignin, milled wood and humic substances, as well as various xenobiotic compounds [108,118].

In the crystal structure of the isoform MnP 1 produced by *P. chrysosporium* were identified in the vicinity of one of the two heme propionates three acidic amino acid residues (two Glu, E35 and E39; and one Asp, D179) that are responsible for the Mn^2+^-binding site [119]. This Mn-binding site is crucial for the hexacoordination of the Mn^2+^ ion, in this way supporting fast electron transfer to heme (Compound-I→Cpd-II) and ferryl iron (Compound-II→resting state MnP; Figure 4). Finally, to end the cycle of the enzyme, two Mn^2+^ ions are oxidized into two Mn^3+^ ions; these ions diffuse out from the binding site as chelate complexes formed with dicarboxylic acid anions, such as malonate or oxalate (which are synthetized by the fungi in high levels) [104,120].

### 3.5. Versatile Peroxidases

Versatile peroxidases (VPLs, EC 1.11.1.16, Reactive Black 5:H_2_O_2_ oxidoreductase) have catalytic properties from the enzymes LiP and MnP. VPLs are considered hybrids of both enzymes, with two substrate-binding sites, one for aromatic substrates and one for Mn^2+^ [107,112]. Consequently, they can oxidize Mn^2+^ into Mn^3+^, and moreover, they can oxidize phenolic and non-phenolic substrates in the absence of Mn^2+^ [14,107].

Versatile peroxidases were first described in *Pleurotus eryngii* [112,121] and were also reported in *Bjerkandera* spp. [122], *Trametes* [123] and *P. ostreatus* [107]. The catalytic cycle of *P. eryngii* VPL is a combination of the LiP- and MnP-specific activities (Figure 5); the LiP-characteristic exposed tryptophan residue is found in VPLs (Trp-164) together with the three MnP-characteristic acidic amino acid residues (2 Glu, 1 Asp) involved in Mn^2+^ binding [102,117].

Compared with LiPs, versatile peroxidases from *Pleurotus* strains are preferrable from the point of view of biotechnological applications since no mediator is required for the oxidation of recalcitrant substrates.

### 3.6. Dye-Decolorizing Peroxidases

DyP-type peroxidases (EC 1.11.1.19) belong to the largest superfamilies of heme peroxidases; however, these enzymes are unrelated in sequence and structure to class I and class II peroxidases [99,103,124]. These peroxidases are currently classified as the DyP-type peroxidase family and they show only little sequence similarity (0.5–5.0%) to classic fungal peroxidases since they lack the typical heme-binding region which is conserved in the whole plant peroxidase superfamily (one proximal histidine, one distal histidine, one essential arginine) [30,125]. Additionally, aspartic acid and arginine are conserved in the H_2_O_2_-binding site of most DyP-type peroxidases, while class II fungal peroxidases have histidine and arginine at this site. Additionally, DyP-type peroxidases are further sub-classified into the phylogenetically distinct classes A, B, C and D [125].

The first suggestions of the existence of this kind of peroxidase were made by Kim et al. [126] after the screening of microorganisms for the decolorization of xenobiotic dyes. A fungal strain of *Geotrichum candidum* decolorized 18 types of reactive, acidic and dispersive dyes. Later, Kim and Shoda [127] purified and characterized the enzyme and named it DyP. It is a glycosylated heme protein (17% sugars) with a pI of 3.8 and a molecular mass of 60 KDa.

Later, the presence of this family of peroxidases have been described in WRF such as *Termitomyces albuminosus* [128], *P. ostreatus* [129], *Marasmius scorodonius* [130], *Auricularia auricula-judae*, *Exidia glandulosa*, *Mycena epipterygia* [131], *Irpex lacteus* [132], *Funalia trogii* [132], *T. versicolor* [133] and *Pleurotus sapidus* [134]. Moreover, it has been described in a variety of organisms, such as archaea, bacteria, fungi and higher eukaryotes, suggesting divergent physiological roles [124]. Some authors suggest that some bacterial variants are associated with lignin degradation [124], while in basidiomycetes, DyP peroxidases can use H_2_O_2_ to detoxify different groups of azo- and anthraquinone dyes; however, the physiological role of these enzymes is still unclear [107,131].

The catalytic mechanism of class II fungal peroxidases occurs via the formation of compound I, which is formed by a reaction between H_2_O_2_ and the Fe(III) resting state of the enzyme. Regarding the mechanism of DyP peroxidases, it is generally accepted that compound I is also formed; however, the details of the catalytic cycle have not been elucidated yet [124]. LRET between the heme cofactor and the surface of DyPs has been suggested as a potential mechanism since these peroxidases are capable of oxidizing substrates that are too large to fit into the active site of the enzyme. The LRET pathway to the surface of a DyP from *Auricularia auricula-judae* has been identified [124]. Further structural studies are needed to uncover the molecular details of how DyPs catalyze oxidation [131].

The distinctive characteristics of DyPs from WRF is their ability to degrade anthraquinone dyes; nevertheless, such synthetic dyes are not true substrates because they are manmade compounds. This suggests that natural anthraquinone compounds such as alizarin produced by plants must be their natural substrate. The probable physiological role of DyPs in WRF that parasitize trees is to degrade antifungal anthraquinone compounds and accelerate the parasitization of the tree.

DyPs are also highly stable, which is an important prerequisite for many biotechnological applications. For example, Puhse et al. [130] described that purified dimeric MsP1 DyP-type peroxidase from *Marasmius scordonius* was remarkably stable under both high pressure (up to 2500 bar; 250 MPa) and elevated temperatures (up to 70 °C).

### 3.7. Fungal Monooxygenases: The Cytochrome P450 Monooxygenases

Many authors have demonstrated the significant contribution of laccases and peroxidases to the degradation of xenobiotic compounds [16,135,136]. However, there is growing evidence that intracellular enzymes of WRF such as monooxygenases, especially cytochrome P450s (CYP450), are also associated with the degradation of organic pollutants [103,137].

Cytochrome P450s (CYP450) are heme-containing proteins and, based on their characteristics, they are usually referred to as monooxygenases or mixed functional oxidoreductases. These enzymes are one of the largest families of proteins, which are extensively spread in all kingdoms, possessing detoxifying functions [18]. They were discovered in the early 1960s [138] and are characterized for having unique spectral characteristics and for their hemin protein. The ability of reduced CYP450 to form an absorption peak at 450 nm when combined with carbon monoxide is still used for the estimation of CYP450 content.

CYP450s can transfer electrons to oxygen and catalyze the oxidation of various organic compounds. The general reaction for CYP450s is:

RH + NAD(P)H + H^+^ + O_2_ → ROH + NAD(P)^+^ + H_2_O (van den Brink et al. [139])

They can act as terminal monooxygenases catalyzing several reactions, such as carbon hydroxylation, dealkylation, epoxidation, reduction, dehalogenation, deamination, desulfurization and N-oxide reduction, participating in different reactions that contribute to the transport and metabolism of organic substrates [18,137].

A typical system of CYP450 in eukaryotic cells comprises a P450 oxidoreductase and P450 monooxygenase: both are proteins associated with membranes. CYP450 systems are classified into 10 classes according to the proteins participating in the electron transfer. In fungi, classes II, VIII and IX have been identified; class II, being the most common class, includes the CYP450 reductase (CPR) that contains as prosthetic groups flavin adenine dinucleotide (FAD) and flavin mononucleotide (FMN), which deliver two electrons from NAD(P)H to several CYP450 enzymes [137,140].

WRF P450 monooxygenase systems have a wide range of biotechnological applications, especially for the elimination and detoxification of many xenobiotics, such as PAHs [141], pesticides [142], insecticides [143] and PhACs [144].

Using CYP450 inhibition experiments, several authors have provided indirect evidence of their role in bioremediation processes. For example, the involvement of CYP450s in the degradation of pollutants such as pharmaceutical compounds in *T. versicolor* [145] and in *P. ostreatus* [146], of dichlorodiphenyltrichloroethane (DDT) in the basidiomycete fungus *Marasmiellus* sp. [147], of bisphenol A (BPA) in *P. sordida* [148] and *P. chrysosporium* [149], of antibiotics in *Pycnoporus sanguineus* and *P. chrysosporium* [150] and of polychlorinated biphenyl (PCB) in *Phlebia acanthocystis* [151] has been demonstrated.

Additionally, direct evidence for the ability of P450s from WRF to remove organic pollutants has also been provided. Syed et al. [152] demonstrated the oxidation of pyrene by the P450s of *P. chrysosporium*. Ning and Wang [153] demonstrated the oxidation of pentachlorophenol (PCP) by *P. chrysosporium* P450 enzymes. Sakai et al. [154] characterized a cytochrome P450 (CYP505D6) from *P. chrysosporium* and observed that naphthalene was transformed into 1-naphthol and 1,3-dihydroxynaphthalene.

Taking into account all these results, the important role of CYP450 in the degradation of several pollutants is obvious; however, further research will be needed for the application of this in the fungal bioremediation of polluted ecosystems and the treatment of industrial wastewater.

Other intracellular or cell-bound enzymes associated with the degradation of xenobiotics are shown in Table 1. Nitroreductases are widespread among fungi and reduce 2,4,6-trinitrotoluene (TNT) into hydroxylamino- and amino-dinitrotoluenes, which are excreted and may undergo various further enzymatic and non-enzymatic degradation steps [124,155].

Quinone reductases of white rot and brown rot basidiomycetes are involved in quinone redox cycling, which initiates extracellular Fenton reactions that lead to the production of hydroxyl radicals and result in the spontaneous hydroxylation and dehalogenation of aromatic and aliphatic pollutants. Quinone reductases also reduce the quinones that result from the transformation of pollutants by extracellular oxidoreductases. This detoxifies quinones and converts them back into substrates of extracellular oxidoreductases, as demonstrated in the chlorophenol metabolism in ligninolytic basidiomycetes.

### 3.8. Biotechnological Applications of Ligninolytic Enzymes

Ligninolytic enzymes such as laccases, MnPs, LiPs and VPLs exhibit a wide range of substrate specificity and therefore can degrade a broad range of xenobiotic compounds, including synthetic dyes, chlorinated phenolics, pesticides, polycyclic aromatic hydrocarbons and chlorophenols, as well as nitroaromatics and explosives [103,156]. Moreover, they can be involved in the degradation of emerging pollutants such as pharmaceutical compounds, flame retardants and PFASs.

Consequently, laccases, MnPs and LiPs can have many biotechnological applications, with particular emphasis on the bioremediation and decoloration of industrial wastewater such as that from the textile, distillery, pulp and paper industries and many other types of wastewater [157].

Specific laccase enzymes are highly suitable for various industrial applications, such as the detoxification of agricultural byproducts, including olive mill waste or coffee pulp; biobleaching of synthetic dyes in the pulp and paper industries; in food processing; the production of biofuels; in synthetic chemistry; in food processing; pharmaceutical and nano-biotechnological applications; in cosmetics and as biosensors [16,63,158]. In addition to substrate oxidation, laccases can also immobilize soil pollutants by binding them to soil humic substances [16] (Table 1).

MnPs have been also applied to ethanol production, delignification, the degradation of phenolic and non-phenolic compounds, pesticides and pharmaceutical compounds [155,159].

**Table 1 jof-10-00167-t001:** Ligninolytic enzymes produced by white rot fungi, their localization, reaction mechanisms and applications.

Enzyme	Localization	Reaction Mechanism, Substrates	Applications	Reference
**Laccases** (EC 1.10.3.2) benzene diol: oxygen oxidoreductase	Mainly extracellular Some intracellular	O_2_-dependent - Blue copper oxidases - Oxidation of aromatic substrates concomitantly with the four-electron reduction of molecular oxygen into water - Redox potential close to 800 mV - Oxidation of phenolic aromatic compounds and non-phenolic compounds in the presence of redox mediators	- Degradation of a broad range of xenobiotic compounds, including chlorinated phenolics, synthetic dyes, pesticides, polycyclic aromatic hydrocarbons, pharmaceutically active compounds - Bleach kraft pulp - Detoxification of agricultural byproducts, including olive mill wastes or coffee pulp - Commercialized applications of laccase such as: the bleaching of denim with the products Denilite II (genetically engineered and optimized laccase from *T. villosa*, heterologously produced in *Aspergillus oryzae* by novozymes A/S) and ECOSTONE LCC 10 (genetically modified laccase from Thielavia sp. by Darmstadt). treatment of cork stoppers to eliminate aromatic compounds in wine, which produce a bad taste in wine (product Suberase, distributed by the company Novozymes A/S). - Chemical synthesis, biofuel cells Other innovative applications of laccases are: - The immobilization of soil pollutants by coupling to soil humic substances - Chemical synthesis - Biofuel cells - Biosensors - Pharmaceutical and nanobiotechnological applications	Baldrian [16] Giardina et al. [58] Sirim et al. [62]
Tyrosinases Grouped into two enzyme subclasses **Oxidase** (EC 1.10.3.1) and **monooxygenase** (EC 1.14.18.1)	Mainly intracellular or Cell-wall-associated	- O_2_-dependent hydroxylation - Oxidation of ortho-substituted diphenols into catechols (catecholase activity) and hydroxylation of para-substituted phenols (cresolase activity).	- Oxidation of various phenols, including those that are highly chlorinated - Immobilized tyrosinase from *A. bisporus* has been used in biosensors for the detection of phenolic compounds such as the organophosphorous pesticide dichlorvos	Hofrichter and Ulrich [103]
**Lignin peroxidases** (EC.1.11.1.14) diarylpropane: oxygen, H_2_O_2_, oxidoreductase	Extracellular	- H_2_O_2_-dependent - Redox potential of 1400 mV - Oxidize non-phenolic aromatic lignin moieties and similar compounds such as nonphenolic β-O-4 linkage type arylglycerol-aryl ethers in the presence of the redox mediator veratryl alcohol - Oxidation of the substrates occurs via the long-range electron transfer (LRET) pathway attributed to Trp171, which is linked to the heme group	- Degradation of pollutants with high redox potentials PAHs, chlorophenols, nitroaromatics and explosives - Delignification of feedstock for bioethanol production - Coal depolymerization - Treatment of effluents from pulp and paper industry - Production of new beauty lightening products	Husain et al. [160] Biko et al. [52] Chowdhary et al. [104] Singh et al. [14]
**Manganese peroxidases** (EC 1.11.1.13; Mn(II): H_2_O_2_ oxidoreductase)	Extracellular	- H_2_O_2_-dependent - One-electron oxidation of Mn^2+^ into Mn^3+^, chelated Mn^3+^-mediated oxidation of phenolic structures such as lignin, milled wood, humic substances, aromatic amines and xenobiotics - Redox potential between 1000 mV and 2000 mV	- Biotechnological applications such as the production of natural aromatic flavor; decoloration of various types of industrial wastewater such as textile, distillery, pulp and paper industry wastewater; de-lignification and degradation of phenolic and non-phenolic compounds; pesticides and pharmaceutical compounds	Hofrichter [105]; Husain et al. [161] Kumar and Arora [156] Bilal et al. [159]
**Versatile peroxidases** (EC 1.11.1.16) Reactive Black 5:H_2_O_2_ oxidoreductase)	Extracellular	- H_2_O_2_-dependent - Share catalytic properties with LiPs and MnPs, with two substrate-binding sites, one exclusive to Mn^2+^ and one for aromatic substrates - sup>- Oxidation of Mn^2+^ into Mn^3+^ and oxidation of phenolic and non-phenolic substrates in the absence of Mn^2+^ - Oxidation of the substrates occurs via the LRET pathway attributed to Trp164, which is linked to the heme group	- Removal of xenobiotic compounds such as phenolic and non-phenolic compounds, pesticides, high-redox dyes and polycyclic aromatic hydrocarbons However, the practical applications of this enzyme are restricted due to their unavailability in high titers	Pérez-Boada et al. [112] Ruiz-Dueñas et al. [101] Barber-Zucker et al. [162]
**DyP-type peroxidases** (EC 1.11.1.19)	Extracellular	- H_2_O_2_-dependent one-electron oxidation of organic compounds such as anthraquinone dyes - LRET between the heme cofactor and the surface of DyPs - Redox potential between 1200 and 1500 mV - Additional hydrolyzing activity - High stability at high pressure and elevated temperatures	- Treatment of wastewater from textile industries - Applications in medicine and food industries due to its capacity to degrade β-carotene	Puhse et al. [130] Colpa et al. [124] Xu et al. [163]
**Cytochrome P450** **Monooxygenases** **(EC 1.14.14.1)**	Cell bound	- Catalyzing several reactions such as carbon hydroxylation, dealkylation, epoxidation, reduction, dehalogenation, deamination, desulfurization and N-oxide reduction - Transport and metabolism of organic substrates	- Elimination and detoxification of many xenobiotics such as PAHs, pesticides, insecticides and PhACs	Young et al. [141] Mori et al. [143]
**Phenol** **2 monooxygenases** (EC 1.14.13.7)	Cell bound	- Incorporation of a single atom from O_2_ into a substrate, with concomitant reduction of the other atom into H_2_O - *Ortho*-hydroxlyation of various (halo) phenols into the corresponding catechols	- Degradation of phenolic compounds from industrial wastewater	Hofricher and Ulrich [103] Harms et al. [155]
**Nitroreductases** (EC: 1.5.1.34)	Cell bound	- NAD(P)H-dependent reduction of nitroaromatics into hydroxylamino and amino(nitro) compounds and of nitro functional groups of N-containing heterocycles - Reduction of TNT into hydroxylamino-dinitrotoluene and amino-dinitrotoluenes - Formation of mononitroso derivatives and ring cleavage products from cyclic nitramine explosives	- Remotion of explosives	Harms et al. [155]
**Quinone** **Reductases** (EC 1.6.99.2)	Cell bound	- NAD(P)H-dependent reduction of quinones - Initiate extracellular Fenton reactions	- Detoxification of quinones	Harms et al. [155]

LiPs can depolymerize synthetic lignin and oxidize diverse xenobiotics, such as synthetic dyes, chlorophenols, PAHs, as well as nitroaromatics and explosives [103] (Table 1). Moreover, diverse biotechnological applications of LiPs have been adopted in various sectors in recent years [104]. LiPs have also been used in the delignification of feedstock for bioethanol production, coal depolymerization, the treatment of effluent from the pulp and paper industry and treatment of hyperpigmentation. For example, LiP isolated from *P. chrysosporium* was used in the decolorization of synthetic melanin and consequently applied in the production of new cosmetic lightening products [104,164,165]. However, the commercial applications of LiPs are restricted due to the scarcity and high cost of commercial LiP preparations. Despite the advances existing in molecular biology techniques, heterologous expression systems secreting high levels of LiPs are not available, and the production of LiPs at a large scale for industrial and biotechnological applications remains a challenge [52]. Additionally, another disadvantage of applying LiPs is the need for H_2_O_2_ in their catalytic cycle; it is a challenge to provide and deliver this compound during the enzymatic treatment of xenobiotics.

The biotechnological applications of DyPs are less broad than the other enzymes discussed in this review. However, due to the extensive capacity of DyPs to degrade lignin, these enzymes are a candidate for the management of lignin produced as waste from the biorefinery, paper and pulp industries. Moreover, DyPs have high decolorization efficiencies for anthraquinone dyes, but also azo- and triarylmethane dyes, having great potential for the treatment of wastewater from textile industries [163]. Moreover, DyPs have been used in food and medicine production processes due to their capacity to degrade β-carotene and whiten foods and beverages containing whey [166]. In spite of the great application potential of this enzyme, the low activity of DyPs and their small pH range limit their practical application. The development of gene recombination and directed evolution technologies could provide a solution for the commercial application of DyPs [166].

## 4. Mechanisms Used by Fungi for the Degradation of Xenobiotics

Among the main mechanisms used by fungi to degrade xenobiotics are biosorption, biodegradation and enzymatic mineralization. These processes can occur simultaneously or separately [167]. Figure 6 shows the mechanisms and enzymes associated with the initial intracellular attack and extracellular oxidation.

The initial pollutant attack may occur extracellularly or intracellularly. The metabolites generated during extracellular pollutant oxidation may be subject to intracellular catabolism or may be bound to soil constituents. The metabolites arising from the initial intracellular attack may be excreted and can then either undergo further extracellular enzymatic reactions or form bound residues through abiotic oxidative coupling [155]. They may also be secreted in the form of conjugates (which usually persist) or may undergo further intracellular catabolism. This may result in mineralization or, again, in metabolite excretion at various oxidation stages if subsequent oxidation is impeded [18].

## 5. Bioremediation by White Rot Fungi

Bioremediation techniques can be applied to treat polluted wastewater, soil, air, surfaces or the ground and sediments, where the presence of pollutants pose a risk to the ecosystem and/or the health of animal and humans [160]. By using bioremediation, the pollutants should be lowered to undetectable or acceptable nontoxic levels.

Bioremediation by fungi, also called mycoremediation, can be carried out in situ or ex situ [160]. The techniques for in situ bioremediation include the treatment of pollutants in the polluted location and can involve natural attenuation, bioaugmentation, bioventing, bioslurping and biosparging [160,168,169]. All these methods imply little or no disturbance to the soil structure and are less expensive compared to ex situ bioremediation techniques. These methods have been successfully applied to the bioremediation of chlorinated-solvent-, dye-, heavy-metal- and hydrocarbon-polluted sites [170,171]. On the other hand, the techniques for ex situ bioremediation implicate removing or excavating the polluted soil, sediments or water and subsequently transporting them to another site for appropriate treatment [169]. For solid matrices, biopiling, composting and land farming are used, whereas bioreactors are used for polluted water [168,169]. The implementation of these techniques depends on whether the pollutant can be collected and transported to facilities for remediation, e.g., designated landfills, large collection tanks and bioreactors or a combination of existing solid waste and wastewater treatment facilities with designated sections. Ex situ mycoremediation enables better control over the environmental parameters and the growth of the fungi. Moreover, the high requirement of oxygen for the fungi can be met using ex situ bioremediation techniques. By using these techniques, genetically modified organisms (GMOs) can be used, since using GMOs is only allowed in closed systems, where they can be deactivated after the biological treatment [172].

Fungal bioremediation using WRF can be conducted using fungal biomass or purified enzymes (free or immobilized). In bioreactors, the fungal biomass can be in pellets or immobilized in different solid support materials [173,174].

The choice of using fungal biomass or an enzymatic treatment for the bioremediation of ecosystems with xenobiotics is dependent on the matrix [23]. To achieve an in situ mycoremediation of solid/hazardous waste, the use of fungal biomass has been more frequently reported in recent years [175,176,177]. However, for aquatic ecosystems, the uses of crude enzymatic extracts or purified ligninolytic enzymes seems to be more suitable to accomplish successful bioremediation [178,179].

For the ex situ treatment of liquid waste containing organic pollutants or the treatment of industrial wastewater in bioreactors, both enzymes (free or immobilized in a variety of carriers) and fungal biomass (in pellets or immobilized in different types of solid carriers materials) have been reported using different bioreactor designs with different modes of operations (batch or continuous conditions).

At present, there is no versatile method that is suitable for all situations or polluted ecosystems; each of the methodologies has advantages and limitations depending on the xenobiotics, the type of matrix and the environmental conditions.

Most recent studies have demonstrated the application of laccases and peroxidases to the treatment of pollutants present in wastewater [180] due to the advantages of their use compared to fungal treatment. Among the advantages are the shorter time for the detoxification of the pollutants, the lower volume of sludge and operation at low and high concentrations of organic pollutants across a wide range of temperatures, pHs and salinity. Moreover, these enzymes are less inhibited by different organic and inorganic compounds which are present in industrial wastewater, which cause toxicity to living organisms [160,179].

At the industrial level, the immobilization of enzymes on a solid support material is the preferred method in order to improve the economic feasibility of the use of enzymes. By using immobilized enzymes, there is an increase in the stability of the enzymes against proteolysis and exposure to extreme conditions in terms of temperature, pH and inhibitors. Moreover, the immobilization of the enzymes provides higher productivity per active unit and a longer shelf life [160].

Different methods for the immobilization of laccases and peroxidases have been reported: chemical bonding, adsorption, chemical aggregation, microencapsulation entrapment and bioaffinity immobilization. Different organic and inorganic compounds were used as the support material (wood chips, dextran, Sephacryl, styrene, maleic anhydride-based polymers, solid glass, silica, alumina, ZnO, CaO, TiO_2_) [168,181].

Currently, the expansion of nanotechnology has allowed the development of novel carriers such as carbon nanotubes (CNTs), which are the most popular nanomaterial for the immobilization of enzymes, including laccases. These nanomaterials are characterized by their ease of preparation, high environmental stability, high mechanical strength and the possibility of depositing more protein on their surface. Graphene oxide (GO) attracts a lot of interest in the scientific community because it is characterized by a high stability in water and a large surface area, on which there are oxygen functional groups [182].

### 5.1. Biodegradation of Synthetic Dyes and Textile Wastewater

Synthetic dyes are extensively used in diverse industries such as printing, textiles and leather and the food, pharmaceutical and cosmetic industries due to their variety in color compared with natural dyes. Moreover, synthetic dyes have several uses due to their easy and cost-effective synthesis and high stability regardless of temperature, light, detergent or microbial attack [183,184].

A diversity of textile dyestuffs (approximately 10,000 synthetic dyes; 8 × 10^7^ metric tons) are produced worldwide every year [185]. The textile industry consumes ~75% of the dyes produced. Textile industries are generally established in developing countries such as Bangladesh, India and Sri Lanka, where they improve the employment capacity and contribute to the development of the economy [184]. Nevertheless, due to the limited existence of wastewater treatment systems, these countries often release large amounts of partially or totally untreated dye effluents, resulting in huge amounts of environmental pollution [184]. Moreover, throughout the dyeing process, 10 to 20% of the dyes do not attach to the textile and are released into wastewater. Due to their stability and resistance to degradation, there is no reduction in the concentration of the dyes after conventional treatment of the effluent [186]. Once in the aquatic ecosystems, these compounds reduce the quantity of sunlight accessible to photosynthetic organisms, which causes diminished oxygen levels in the water [187]. Wastewater from these industries is dangerous to the environment and human health since it has been demonstrated that synthetic dyes are toxic, mutagenic and/or carcinogenic. Moreover, these compounds can cause allergies and dermatitis [183,188].

Many investigations have been carried out to find strains of WRF with efficient ligninolytic machinery for the degradation of a variety of dyes with different chemical structures. Researchers have described different WRF species capable of efficiently degrading different types of dyes (azo, anthraquinone, phthalocyanine and triarylmethane): *Bjerkandera* sp., *Irpex lacteus*, *P. eryngii*, *P. ostreatus*, *Pleurotus sajor-caju*, *Polyporus ciliatus*, *Polyporus sanguineus*, *Pycnoporus sanguineus*, *T. versicolor*, *Irpex lacteus*, *Geotrichum candidum*, *Dichomitus squalens*, *Lentinus tigrinus*, *P. chrysosporium*, *P. sordida*, *P. radiata*, *Phlebia brevispora*, *Phlebia tremellosa*, *Ganoderma lucidum*, *Ganoderma weberianum* and *Ganoderma* sp., among others.

Among the species of WRF that have been most studied in the degradation of dyes are *P. ostreatus*, *P. chrysosporium*, *T. versicolor* and *G. lucidum* [18].

In the case of *P. ostreatus*, important studies have been carried out, such as by Pezzella et al. [82], who successfully immobilized crude laccase preparation from *P. ostreatus* on perlite and used it in a fluidized bed recycle reactor for the degradation of Remazol Brilliant Blue R (RBBR). After 4.2 h in a lab-scale reactor according to a continuous regimen, a maximum degradation of 56.1% for this anthraquinone dye was obtained. Such results are important for the future application of such an immobilized biocatalyst to the treatment of textile effluents. Similarly, Dai et al. [189] reported the immobilization of a laccase from *P. ostreatus* on Fe_3_O_4_/SiO_2_ nanoparticles using a coupling method. The phenolic azo dye Procion Red MX-5B was decolorized by almost 100% in one hour. Moreover, the immobilized laccase displayed an excellent storage life and operational stability. The half-life time is close to 50 cycles of decolorization of Procion Red MX-5B with 50% of the initial activity of the immobilized laccase remaining. Skariyachan et al. [190] determined the degradation percentages of the azo dyes nylon blue and cotton yellow and their effluents from Karnataka Silk Industries Corporation (KSIC) and Ramanagar (India). The crude enzymatic extracts obtained after the cultivation of *P. ostreatus* (Jacq.: Fr.) P. Kumm reached high percentages of decolorization (78.10% of nylon blue, 90.81% of cotton yellow, 82.5% of KSIC effluents and 64.88% of Ramanagar effluents after 15 days of treatment). Zhuo et al. [77] reported the dye decolorization efficiency of three *P. ostreatus* HAUCC 162 laccase isoenzymes heterologously expressed in *Pichia pastoris* (currently named *Komagataella phaffii*). Among the recombinant laccases, LAC 6 showed the highest ability to remove Remazol Brilliant Blue R (RBBR), Bromophenol blue (BB), Methyl orange (MO) and Malachite green (MG), with values between 73.1% and 91.5% after 24 h of incubation, suggesting that this laccase enzyme can be used for the treatment of textile industry effluents. George et al. [191] utilized porous cross-linked enzyme aggregates (CLEAs) of *P. ostreatus* laccase for the decolorization and detoxification of triarylmethane and azo dyes, reactive blue 2 (RB) and malachite green (MG). The CLEAs of the laccase decolorized 500 ppm of MG and RB with 98.12 and 58.33% efficiency after 120 min, at a pH 5.0 and 50 °C, without a mediator. The reusability potential of the CLEAs was evaluated in batches for 10 cycles of dye decolorization. This immobilized enzyme could successfully remove dyes from aqueous solutions and demonstrated important detoxification for plants (*Triticum aestivum* and *Phaseolus mungo*) and plant-growth-promoting rhizobacteria (*Azospirillum brasilense*, *Bacillus megaterium*, *Rhizobium leguminosarum*, *Bacillus subtilis* and *Pseudomonas fluorescens*). The use of porous CLEAs of laccase can become a suitable alternative for the decolorization and detoxification of dyeing wastewater in future.

Another species of WRF that was widely studied for the degradation of synthetic dyes and that is also considered a model for understanding ligninolytic enzyme production systems is *P. chrysosporium*. This fungus is found in temperate forests throughout the world, and it is a decomposer of both softwoods and hardwoods [18].

Freire Andrade et al. [192] demonstrated the capacity of *P. chrysosporium* to decolorize the azo dye Congo Red in a batch reactor reaching 97% of removal with the addition of glucose as a co-substrate. Rani et al. [193] studied the biodegradation and also the detoxification of various dyes, such as Nigrosin, Basic Fuchsin, Malachite Green and a mixture of dyes, reaching high percentages of decolorization and detoxification after 6 days (between 78.4% and 90.15%; see Table 2). Li et al. [194] developed an interesting approach to dye decolorization. They co-immobilized *P. chrysosporium* cells and cross-linked enzyme aggregates (combi-CLEAs) prepared from *T. versicolor* onto Ca alginate gel particles. The combi-CLEA particles improved the degradation of the textile effluents (Acid Violet 7 (from 45.2% to 93.4%) and Basic Fuchsin (from 12.1% to 67.9%). Using the fungi *P. chrysosporium*, Wanderley et al. [195] reported the transformation of the azo dye Congo Red in two sequential batch bioreactors operated in cycles of 24 h and 48 h. In this case, laccase enzymes were produced in higher amounts. Sierra-Solache et al. [196] reported on the utilization of the *P. chrysosporium* strain EMIM 5 in global wastewater treatment of textile effluents based on bioremediation and ultrafiltration processes. *P. chrysosporium* were immobilized onto spheres of alginate–polyvinyl alcohol–graphene and used in aerated bioreactors. The wastewater samples were treated using encapsulated and free fungal cells within stirred and airlift bioreactors with the subsequent ultrafiltration process. The maximum efficiency was reached by the reactor with stirring treatments, displaying 70% and 80% COD and coloration removal, respectively, whereas the treatment in the airlift reactor showed 50–70% removal under the same operational conditions. The combined treatment using the stirred bioreactor followed by the ultrafiltration process yields 90% and 100% COD and coloration removal, respectively, making it possible to reuse the water recovered in the textile industry.

Oliveira Santos et al. [197] evaluated the use of immobilized *P. chrysosporium* for the treatment of synthetic effluent containing indigo carmine (20 mg·L^−1^) in a semi-batch reactor. After the treatment in the bioreactor, the concentration of indigo dye decreased to 3.35 ± 1.99 mg·L^−1^, indicating the efficiency of the bioreactor system. Pereira de Pereira-Almeida et al. [198] evaluated the capacity of the *P. chrysosporium* strain ME-446 to decolorize and detoxify three azo dyes (Direct Yellow 27 (DY27), Reactive Black 5 (RB5) and Reactive Red 120 (RR120). The decolorization efficiencies reached by the fungal biomass of *P. chrysosporium* were 82% for DY27, 89% for RB5 and 94% for RR120 at a concentration of 50 mg·L^−1^ after 10 days of treatment. The authors concluded that the removal of dyes was achieved due to their adsorption onto the fungal mycelium as well as biodegradation. A phytotoxicity test using *Lactuca sativa* seeds indicated that the fungal treatment reduces the toxicity of RB5 and DY27. The results support the suitability of this strain for the development of biological treatment systems.

*G. lucidum* is one of the most important WRF and is broadly distributed. *G. lucidum* and other species of this genus possess medicinal properties [199].

Some authors who studied different *Ganoderma* strains reported the production of lignin-modifying enzymes with a capacity to degrade recalcitrant synthetic dyes and other xenobiotics [200,201,202]. Selvakumar et al. [203] reported the utilization of a strain of *G. lucidum* for the biological treatment of a textile effluent using a batch reactor. After the optimization of the process, the decolorization and COD were 81.4% and 90.3, respectively. A correlation between the laccase activity and the decolorization process was observed.

Ma et al. [204] reported the decolorization and detoxification of high concentrations of the sulfonated azo dye Reactive Orange 16 using *G. lucidum* En3. After 96 h of incubation, the rates of decolorization under optimized carbon and nitrogen conditions were 98.2% and 74.6% for concentrations of 1000 and 8000 mg·L^−1^, respectively. Furthermore, they tested the decolorization of simulated wastewater containing Reactive Orange 16, and the decolorization percentages oscillated between 73.2% and 89.5% after 10 days of treatment.

Rainer et al. [205] detailed the decolorization of the anthraquinone dye Remazol Brilliant Blue (RBBR) using *G. lucidum* EF33 immobilized onto a solid bleached sulfate paperboard coated with polyethylene terephthalate. The bioabsorbent successfully decolorized the RBBR (96% of removal) and wastewater containing this synthetic dye; additionally, it was used and reused for 30 days. The authors concluded that this combination of fungal strains and absorbent is an efficient and low-cost alternative for the treatment of wastewater containing synthetic dyes.

The *Ganoderma* strains studied are producers of laccase enzymes in higher amounts. Consequently, different authors have reported the application of the laccase enzymes produced by various strains of *Ganoderma* genus. For example, Palazzolo et al. [202] reported the biotechnological application and characterization of the laccase produced by the *G. lucidum* E47 strain in solid-state fermentation (SSF) using rice straw, husks and bran. The laccase enzyme decolorized efficiently the synthetic dyes xanthene, azo and triarylmethane, with bromocresol green and bromocresol purple being the dyes with higher percentages of decolorization. Furthermore, the dye bromocresol green was decolorized completely in a batch recirculation flow mini-reactor of 0.5 L, simulating a conventional wastewater treatment process. Himanshu et al. [206] described the decolorization of malachite green using purified laccase from *G. lucidum* MTCC-1039 produced under SSF. After optimization using the Box–Behnken design, 72% decolorization was obtained in 1 h at a pH of 8 and at 50 °C and the maximum after 3 h at 91% in the presence of a redox mediator. Additionally, according to a test using the seeds of *Cyamopsis tetragonoloba* (dicot) and *Pennisetum glaucun* (monocot), the phytotoxicity was reduced after the enzymatic treatment, demonstrating its efficiency.

Another *Ganoderma* species applied to the degradation of synthetic dyes and textile wastewater is *G. weberianum*. Torres-Farradá et al. [207] reported the use of fungal biomass of *G. weberianum* B-18 immobilized in sugarcane bagasse in a packed-bed bioreactor for the decolorization of the anthraquinone dye RBBR and textile wastewater containing the dyes Cibacron violet W-HB and Bezanthrene orange GR. The detoxification of the dyes after the treatment was tested using the seeds of *Oryza sativa*, demonstrating a reduction in the toxicity of the dyes and the industrial textile effluents after the biological treatment. The authors concluded that *G. weberianum* B-18 immobilized in sugarcane bagasse appears to be a suitable system for the further development of an efficient bioprocess for large-scale treatment of dye-containing wastewater.

### 5.2. Biodegradation of Polycyclic Aromatic Hydrocarbons

Polycyclic aromatic hydrocarbons (PAHs) are fused-ring aromatic compounds and a heterogeneous group of greatly toxic organic pollutants. They originate from natural and anthropogenic processes. Their principal source is the incomplete pyrolysis of fossil organic materials or fuels such as oil, petroleum gas, coal and wood or the processes related to the petrochemical industries [208]. PAHs have been recognized for their mutagenic and carcinogenic effects, with considerable consequences on human and environmental health [209].

The species of WRF that have been most extensively studied for the degradation of diverse types of PAHs and used for the in situ bioremediation of polluted soils are *P. ostreatus* and *P. chrysosporium*.

Ding et al. [210] reported the biosorption and biodegradation of pyrene and phenanthrene by *P. chrysosporium* and found that both processes contributed to the elimination of PAHs using live *P. chrysosporium* in water. Under carbon-rich and nitrogen-limiting conditions, after 60 days, respectively, 99.55% and 92.77% of the phenanthrene and 99.47% and 83.97% of the pyrene, was degraded. Wang et al. [211] evaluated the degradation of pyrene in different soils. The highest level of pyrene degradation (66.20%) was observed in non-sterilized unaged soils after 19 days. The study showed that a higher soil organic matter content negatively affected the pyrene degradation. The authors also reported the use of tourmaline combined with *P. chrysosporium* to bioremediate soils polluted with PAHs and organochlorine pesticides (OCPs). After 60 days of treatment, the degradation of the PAHs and OCPs was 53.2% and 43.5%, respectively.

Pozdnyakova et al. [6] investigated the degradation of two PAHs using *P. ostreatus* D1 (fluorene and fluoranthene) in Kirk’s medium with high laccase production. They described the intermediate metabolites that were formed during the degradation of these PAHs, such as 9-fluorenone and phthalic acid. They suggested that both intracellular and extracellular laccase have an important role in the initial stages of PAH metabolism, while versatile peroxidase is essential for the oxidation of the formed metabolites. A scheme of fluorene degradation using *P. ostreatus* D1 is proposed. Elhusseiny et al. [212], using a strain of *P. ostreatus*, investigated the biodegradation and expression levels of laccase genes associated with the degradation of naphthalene, anthracene and 1,10-phenanthroline under different conditions. The naphthalene degradation, at 100%, was higher compared to that of anthracene (93.69%) and 1,10-phenanthroline (92.00%), after 5 days of incubation for the naphthalene and after 14 days for the rest of the PAHs analyzed. Based on the detected metabolites, the metabolic pathway of naphthalene degradation for this fungus was elucidated.

### 5.3. Biodegradation of Pharmaceutically Active Compounds (PhACs)

Among the different pharmaceutical substances, pharmaceutically active compounds (PhACs) are xenobiotic-based components that enter the environment and remain active as unmetabolized parent compounds or as pharmacologically active metabolites [18,213]. Among the PhACs are analgesics, antibiotics, anticonvulsants, beta-blockers, steroids, hormones and X-ray contrast media. The principal routes through which PhACs are released into the environment are patient excretion, direct release into the wastewater system from hospitals, disposal through toilets and sinks and the irrigation of soils with untreated wastewater [214]. Such compounds are not totally eliminated in the conventional wastewater treatment, and residual concentrations are discharged into superficial and ground water [215]. The presence of PhACs in the environment is a growing concern due to their risks to the environment and human health due to their bioaccumulation capacity, persistence, water solubility and toxicity [169]. These substances can cause analgesic tolerance in humans or antibiotic resistance in bacteria and also undesirable effects in animals and humans, such as abnormal protein synthesis, alterations in gene expression, alterations in the sex ratio and diminished fertility, even at very low concentrations [18,23,169]. According to the European Union (EU), about 3000 different substances classified as PhACs were found downstream [212]. Nowadays, the EU specifies quality standards and priority pollutants, but there are no concentration limits in terms of the pharmaceutical compounds in effluents [216].

WRF and their enzymes have the potential to degrade a range of pharmaceuticals, from analgesics [215] to antibiotics and antidepressants [217], beta-blockers [218], anticancer drugs [219] and anti-inflammatory drugs [220].

The most investigated species is *T. versicolor*; however, a variety of fungal strains have also been reported to be able to degrade various PhACs. Marco-Urrea et al. [215] studied the degradation of carbamazepine, ibuprofen and clofibric acid using strains of *T. versicolor*, *I. lacteus*, *G. lucidum* and *P.chrysosporium*, demonstrating that all four strains degraded ibuprofen after 7 days of incubation. Nevertheless, the more recalcitrant carbamazepine and clofibric acid were only degraded by *T. versicolor*. Marco-Urrea et al. [145] reported the removal of 94% of diclofenac in just one hour using a strain of *T. versicolor*. The cytochrome P450 system was the enzyme associated with the initial degradation of diclofenac.

*T. versicolor* has been proposed as an efficient fungus for purifying hospital wastewater. Cruz-Morato et al. [220] reported the use of *T. versicolor* in a bioreactor for the treatment of hospital wastewater under sterile and non-sterile conditions. Of the 51 PhACs detected, 46 of them were degraded partially or completely, with a significant decrease in toxicity. Anti-inflammatory drugs and analgesics such as acetaminophen, naproxen, ibuprofen, phenazone and diclofenac, which were found in elevated concentrations (ranging between 10 and 100 μg·L^−1^) in the hospital effluents, were decreased by more than 80% in 24 h. The removal rates of antibiotics such as metronidazole, trimethoprim, erythromycin and sulfamethoxazole showed variations ranging from 26 to 100% compared to the analgesics. The authors suggest that treatment with WRF is a suitable alternative for the management of wastewater containing pharmaceutical compounds.

Different studies used other WRF species for the degradation of PhACs. Bilal et al. [221] investigated the degradation of pharmaceutical compounds such as sulfamethoxazole, acetaminophen, carbamazepine and the plastic additive bisphenol A by the *Bjerkandera* strain TBB-03. Complete removal of the bisphenol A and acetaminophen was obtained after 2 h of incubation. Carbamazepine and sulfamethoxazole were not well degraded. Jureczko et al. [218] studied the degradation of bleomycin and vincristine (anticancer drugs) by *Fomes fomentarius*, *Hypholoma fasciculare* and also *T. versicolor*, obtaining a high removal efficiency for vincristine (>94%). However, bleomycin was very recalcitrant to degradation, with only 36% removal using *T. versicolor*. Nevertheless, the authors suggested that considering the comparably low concentrations of pharmaceuticals in actual wastewater, WRF could be efficiently applied to their degradation.

An interesting approach to the degradation of PhACs is the use of fungal consortia. For example, a consortium of the edible fungus *Laetiporus sulphureus* and the WRF *Ganoderma applanatum* removed 99.5% of all compounds in a mixture of anti-inflammatory drugs (diclofenac, celecoxib and ibuprofen) after 72 h of incubation [222]. In contrast, the degradation percentage was lower (66–92%) when using only one strain.

### 5.4. Biodegradation of Per- and Polyfluoroalkyl Substances

Per- and polyfluoroalkyl substances (PFASs) are a group of synthetic organic compounds containing at least one F atom and have diverse chemical and physical characteristics. PFASs are highly stable organic compounds comprising perfluorocarbons, perfluoroalkyl acids (PFAAs), perfluoroalkane sulfonyl fluorides (PASFs), perfluoroalkane sulfonamides (PASAs), perfluoroalkyl iodides (PFAIs) and perfluoroalkyl aldehydes [223,224]. They are considered “forever chemicals” due to their recalcitrance and the fact they accumulate in living organisms.

Over the past 70 years, PFASs have been produced and used in diverse industrial applications [225,226]. They are commonly used in non-stick coatings for cookware, as surfactants in the production of fluoropolymers, metal coatings, water-repellent coatings for packaging and clothes, as well as in fire-fighting foams [224,227]. Many industrial and commercial products contain PFASs, which encompass more than 4700 different compounds [228,229]. Although their usage has been restricted for the last decade, PFASs are still commonly detected in the environment (drinking water, aquatic ecosystems and soil), in wildlife and in humans [230]. The main sources of PFAS release are industrial and municipal wastewater treatment plants [231] and municipal landfills [232], due to the inefficient technologies for the removal of these compounds.

Once in the environment, soluble PFASs can be transported over long distances through water, and volatile PFASs can be transported by air, a risk for human exposure through indoor air and dust. It has been reported that indoor dust contains perfluorooctane sulfonic acid (PFOS), perfluorooctanoic acid (PFOA) and perfluorohexane sulfonate (PFHxS) at different concentrations. Ramhøj et al. [233] reported that concentrations above 3 and 10 mg/kg^−1^ of body weight per day are harmful to the liver and thyroid, respectively. Humans are exposed to these toxic compounds through consumer products, drinking water, food and indoor air/dust. PFASs have been found in the blood of industrial workers [224] and in breast milk [234]. The exposure of humans to PFASs has been linked to various negative health effects, such as reproductive and developmental deficits, immunotoxicity, neurotoxicity, pancreatic tumors, hepatomegaly and hepatic peroxisome proliferation [230,235]. In animals, toxicity tests demonstrated compromised postnatal survival and growth deficits [226]. PFOA and PFOS are the most detected PFASs in aquatic systems and drinking water worldwide. Since these compounds are persistent in the environment and have negative effects in animals and humans, strict regulations were established, together with actions to decrease the amount of PFASs and their precursors that end up in the environment [223,236,237].

Several methods, including adsorption, filtration, chemical oxidation and soil washing, have been developed for the removal of PFASs from waste streams or polluted environments. However, although these technologies have shown promising outcomes in laboratory studies, their cost effectiveness, field applicability and feasibility have not been demonstrated yet [238]. Therefore, innovative treatment technologies need to be explored for the in situ remediation of contaminated water and soil. Due to the potential of WRF to degrade a wide range of recalcitrant compounds, fungal bioremediation might be an alternative for the removal of PFASs from the environment. Surprisingly, there are very few studies examining their ability to degrade PFASs [239].

Luo et al. [240] reported the degradation of PFOA using laccase from a strain of *P. ostreatus*. They obtained 50% degradation after 157 days in the presence of 1-hydroxybenzotriazole as a redox mediator. Moreover, they obtained 40% degradation after 140 days in the presence of soybean meal and laccase in a soil slurry.

Tseng et al. [241] explored the potential of *P. chrysosporium*, *Aspergillus niger* and five fungal strains isolated from a contaminated site, as well as bacterial strains, to degrade FTOH, PFOA and PFOS. They reported a 50% transformation of 6:2 FTOH and a 70% transformation of 8:2 FTOH using *P. chrysosporium* over 28 days. No degradation by *A. niger* was detected. Two of the fungal isolates achieved 20% transformation of the PFOS over 14 days and 28 days, respectively. In contrast, no transformation of PFASs by the bacteria was detected, demonstrating the potential of fungi to degrade these recalcitrant compounds.

Merino et al. [242] described fungal biotransformation experiments using FTOH (C_6_F_13_CH_2_CH_2_OH); when this compound is discharged into the environment, it can be transformed into PFCA and other polyfluoroalkyl substances by means of physicochemical and biological processes. They investigated the fungal transformation of FTOH using the WRF *Gloephyllum trabeum* and *T. versicolor* and six fungal isolates (closely related to the genera *Fusarium*, *Aspergillus* and *Penicillium*) from a location contaminated with PFASs. They detected the 6:2 transformation of FTOH by *G. trabeum* and *T. versicolor* into, respectively, nine and six quantifiable transformation products. All the fungal isolates achieved the transformation of 6:2 FTOH into 5–9 quantifiable transformation products during the 28 days of the experiment under the tested conditions at different molar removals. They also reported the tolerance of the fungal isolates in the presence of 100 or 1000 mg·L^−1^ of perfluorooctanoic acid and perfluorooctane sulfonic acid; some isolates displayed growth at increasing concentrations. 

**Table 2 jof-10-00167-t002:** Degradation of xenobiotics by white rot fungi.

White Rot Fungi	Pollutants & Conditions	Removal Rates	References
**Synthetic dyes and textile wastewater**			
** *Pleurotus ostreatus* **	- Anthraquinone Remazol Brilliant Blue R (RBBR) - Crude laccase preparation on perlite fluidized bed recycle reactor	- 56.1% after 4.2 h in a continuous lab-scale reactor	Pezzella et al. [82]
	- Phenolic azo dye Procion Red MX-5B - Immobilization of a laccase onto Fe_3_O_4_/SiO_2_ nanoparticles using a coupling method	- Decolorization percentages almost 100% in one hour	Dai et al. [189]
	- Azo dyes nylon blue and cotton yellow and textile effluents from Karnataka Silk Industries Corporation (KSIC) and Ramanagar (India) - Crude enzymatic extracts	- 78.10% of nylon blue - 90.81% of cotton yellow - 82.5% of KSIC effluents - 64.88% of Ramanagar effluents after 15 days of treatment	Skariyachan et al. [190]
	- Remazol Brilliant Blue R (RBBR), Bromophenol blue (BB), Methyl orange (MO) and malachite green (MG) - Decolorization efficiency of three laccase isoenzymes heterologously expressed in *Pichia pastoris* (currently named *Komagataella phaffii*)	- Decolorization percentages between 73.1% and 91.5% after 24 h of incubation	Zhuo et al. [77]
	- Dyes reactive blue 2 (RB) and malachite green (MG) (500 ppm) - Decolorization and detoxification using porous cross-linked enzyme aggregates (CLEAs) of laccase	- Between 58.33 and 98.12% efficiency after 120 min, at a pH 5.0 and 50 °C, without a mediator - Demonstrated the reusability potential of CLEAs after 10 cycles of dye decolorization. - Demonstration of the reduction in toxicity of RB and MG after the enzymatic treatment by using plants (*Triticum aestivum* and *Phaseolus mungo*) and bacteria (*Azospirillum brasilense*, *Bacillus megaterium*, *Rhizobium leguminosarum*, *Bacillus subtilis* and *Pseudomonas fluorescens*)	George et al. [191]
** *Phanerochaete chrysosporium* **	- Azo dye Congo Red in a batch reactor	- 97%	Freire Andrade et al. [192]
	- Detoxification of Nigrosin, Basic Fuchsin, Malachite Green and a mixture	- 78.4% and 90.15%	Rani et al. [193]
	- Textile effluents containing Acid Violet 7 and Basic Fuchsin - Co-immobilized *P. chrysosporium* cells and cross-linked enzyme aggregates (combi-CLEAs) prepared from *T. versicolor* onto Ca alginate gel particles	- Acid Violet: 79.34% - Basic Fuchsin (from 12.1% to 67.9%)	Li et al. [194]
	- Azo dye Congo Red - Two sequential batch bioreactors operating in cycles of 24 h and 48 h	- 100%	Wanderley et al. [195]
	- Textile effluents - Immobilization of *P. chrysosporium* cells onto spheres of alginate–polyvinyl alcohol–graphene - Treatment of textile wastewater using immobilized and free fungal cells within stirred and airlift bioreactors followed by an ultrafiltration process	- 80% color removal with bioreactor with stirring treatments - 50–70% with an airlift reactor - 90% COD and 100% color removal with combined treatment using a stirred bioreactor followed by an ultrafiltration process	Sierra-Solanche et al. [196]
	- Indigo carmine (20 mg/L) - Immobilized *P. chrysosporium* in a semi-batch reactor	- Reduction to 3.35 ± 1.99 mg/L	Oliveira Santos et al. [197]
	- Direct Yellow 27 (DY27) - Reactive Black 5 (RB5) - Reactive Red 120 (RR120) - Phytotoxicity test using *Lactuca sativa* seeds - Treatment with fungal biomass over 10 days	Decolorization percentages: - 82% for DY27 - 89% for RB5 - 94% for RR120 - Removal of dyes achieved through adsorption onto the fungal mycelium as well as biodegradation - Reduction in the toxicity of RB5 and DY27	Pereira de Almeida et al. [198]
** *Ganoderma lucidum* **	- Textile effluent - Batch reactor	- 81.4%	Selvakumar et al. [203]
	- Sulfonated azo dye Reactive Orange 16 - Decolorization and detoxification simulated wastewater containing Reactive Orange 16	- 98.2% and 74.6% at concentrations of 1000 and 8000 mg/L - 73.2% and 89.5% after 10 days of treatment	Ma et al. [204]
	- Degradation of xanthene, azo and triarylmethane dyes - Production of laccase in solid-state fermentation using rice straw, husks and bran - Decolorization of bromocresol purple in a batch recirculation flow mini-reactor of 0.5 L simulating a conventional wastewater treatment process	- 100% degradation of bromocresol green after treatment in a mini-reactor	Palazzolo et al. [202]
	- Decolorization of anthraquinone dye Remazol Brilliant Blue (RBBR) - Immobilization of strain *G. lucidum* EF33 immobilized onto solid bleached sulfate paperboard coated with polyethylene terephthalate	- 96% removal of RBBR and wastewater containing this dye - Reuse of bioabsorbent during 30 days	Rainer et al. [205]
	- Decolorization of malachite green using the purified laccase from *G. lucidum* MTCC-1039 produced under solid-state fermentation - Optimization using Box–Behnken design	- 72% decolorization in 1 h at a pH of 8 and 50 °C - 91% decolorization in 3 h in the presence of a redox mediator - Reduction in phytotoxicity using the seeds of *Cyamopsis tetragonoloba* (dicot) and *Pennisetum glaucun* (monocot)	Himanshu et al. [206]
** *Ganoderma weberianum B-18* **	- Anthraquinone dye RBBR - Textile wastewater containing dyes Cibacron violet W-HB and Bezanthrene orange GR - Fungal biomass of *G. weberianum* B-18 immobilized in sugarcane bagasse in a packed-bed bioreactor in semi-continuous conditions	- Decolorization of RBBR in seven addition/extraction cycles (between 72.9% and 87.6%) - More than 64% decolorization of the industrial dyes and effluents - High levels of laccase activity were correlated with high levels of decolorization - Reduction in the toxicity of dyes and industrial textile effluents after the biological treatment	Torres-Farradá et al. [207]
**Polycyclic Aromatic Hydrocarbons**			
** *Phanerochaete chrysosporium* **	- Pyrene - Phenanthrene - Degradation by fungal biomass under carbon-rich or nitrogen-limiting conditions over 60 days	Degradation of phenanthrene: 99.55% under carbon-rich and 92.77% under nitrogen-limiting conditions Degradation of pyrene: 99.47% under carbon-rich and 83.97% under nitrogen-limiting conditions	Ding et al. [210]
	Evaluation of the degradation of pyrene in different soils	- The highest level of pyrene degradation (66.20%) was observed in non-sterilized, unaged soils after 19 days - Higher soil organic matter negatively affected the pyrene degradation - The use of tourmaline combined with *P.chrysosporium* is a suitable alternative for bioremediating soils polluted with PAHs and organochlorine pesticides (OCPs). After 60 days of treatment, the degradation of PAHs and OCP were 53.2% and 43.5%, respectively	Wang et al. [211]
***Pleurotus*** ***ostreatus***	- Degradation of fluoranthene and fluorene by *P. ostreatus* D1 in Kirk’s medium	- Elucidation of the intermediate metabolites formed after the degradation of fluorene and phenanthrene - The authors suggested that both intracellular and extracellular laccase have a principal role in the initial stages of PAH metabolism, while versatile peroxidase is essential for the oxidation of the intermediated metabolites	Pozdnyakova et al. [6]
	- Degradation of naphthalene, anthracene and 1,10-phenanthroline under different conditions - Evaluation of the expression levels of laccase genes associated with the degradation of PAHs	- Degradation percentages: Naphthalene: 100% Anthracene: 93.69% 1,10-phenanthroline: 92.00% after 5 days of incubation for the naphthalene and after 14 days for the rest of PAHs analyzed - Based on the detected metabolites, the metabolic pathway of naphthalene degradation by this fungus was elucidated	Elhusseiny et al. [212]
**Pharmaceutically Active Compounds**			
** *Trametes versicolor* **	- Diclofenac - Cytochrome P450 system was the enzyme associated	- 94%	Marco-Urrea et al. [145]
	- Treatment of hospital wastewater under sterile and non-sterile conditions	- Acetaminophen, naproxen, ibuprofen, phenazone and diclofenac by more than 80% in 24 h - Antibiotics such as metronidazole, trimethoprim, erythromycin and sulfamethoxazole ranging from 26 to 100%.	Cruz-Morato et al. [220]
***Bjerkandera*** **spp. TBB-03**	- Bisphenol A - Acetaminophen	- 100% after 2 h	Bilal et al. [221]
** *Fomes fomentarius* ** ** *Hypholoma fasciculare* ** ** *T. versicolor* **	- Bleomycin and vincristine	- Vincristine (>94%). - Bleomycin 36% removal by *T. versicolor*	Jureczko et al. [218]
**Per- and polyfluoroalkyl substances**			
** *Pleurotus ostreatus* **	- Degradation of PFOA using laccase from a strain of *P. ostreatus*	- 50% degradation after 157 days in the presence of 1-hydroxybenzotriazole as a redox mediator - 40% degradation after 140 days in the presence of soybean meal and laccase in a soil slurry	Luo et al. [240]
** *Phanerochaete chrysosporium* ** ** *Aspergillus niger* ** **Five fungal strains isolated from contaminated site with PFASs**	- Degradation of FTOH, PFOA and PFOS	- 50% transformation of 6:2 FTOH and 70% transformation of 8:2 FTOH by *P. chrysosporium* over 28 days - No degradation by *Aspergillus niger* - 20% degradation by two fungal isolates after 14 and 28 days, respectively.	Tseng et al. [241]
** *Gloephyllum trabeum* ** ** *Trametes versicolor* ** **Six fungal isolates from a location contaminated with PFASs**	- Transformation of 6:2 FTOH	- Transformation of 6:2 FTOH by *G. trabeum* and *T. versicolor* into, respectively, nine and six quantifiable transformation products - All the fungal isolates achieved the transformation of 6:2 FTOH into 5–9 quantifiable transformation products after 28 days - All the fungal isolates tolerated concentrations of 100 or 1000 mg·L^−1^ of perfluorooctanoic acid and perfluorooctane sulfonic acid	Merino et al. [242]

## 6. Current Limitations of Using WRF for the Bioremediation of Polluted Environments and Future Strategies

The intracellular and extracellular enzymatic machineries of WRF make them attractive candidates for bioremediation applications. In spite of their great bioremediation potential, there are certain limitations and challenges that need to be solved to introduce the technology at the industrial and commercial scales. These challenges include the accurate selection of fungal strains, the stability and activity of the ligninolytic enzymes, the environmental factors affecting bioremediation, the design and optimization of reactors and economic feasibility. Here, we discuss some of the current limitations of using WRF for bioremediation and future directions.

Since some xenobiotics such as textile dyes and PAHs, among others, are toxic and can cause growth inhibition in fungi, there is a need to isolate new fungal strains with novel physicochemical characteristics which can tolerate high concentrations of pollutants and, in that way, to obtain better degradation and detoxification rates. An area of intensive research should focus on the isolation of indigenous fungal strains from polluted environments that are already adapted to high concentrations of pollutants and other compounds acting as inhibitors. Additionally, there are a variety of fungal taxa that have not been fully explored yet and that may possibly produce suitable ligninolytic enzymes with high activity and stability under extreme environmental conditions of pH, temperature and ionic strength. For example, fungi from aquatic environments are understudied [243]. The adaptation of marine fungi to high concentrations of heavy metals, high pHs and saline conditions makes these organisms more valuable in extreme conditions than terrestrial fungi. The potential role of the enzymes produced by marine fungi and their biotechnological applications have been described before. Bonugli-Santos [244] detailed the production of ligninolytic enzymes and the decolorization of Remazol Brilliant Blue (RBBR) using marine basidiomycetes (*Tinctoporellus* spp., *Marasmiellus* spp. and *Peniophpra* spp.) isolated from marine sponges.

Some authors suggest that future research should aim at uncovering the extent of fungal species (aquatic or not) that are effective as bioremediation agents. Moreover, it is crucial to decipher the fungal enzymes associated with the degradation of xenobiotics, their metabolic pathways and the toxicity of the degradation intermediates.

Another area of future research should be the exploitation of microbial consortia for the improvement of the degradation of xenobiotics and also a reduction in the time required for degradation.

Furthermore, taking into account advancements in genetic engineering, metabolic engineering and synthetic biology, future research efforts should concentrate on the improvement of WRF strains.

Another issue that limits the practical application of WRF is that most of the existing WRF bioremediation research has been and still is executed under ideal conditions in laboratories. However, for successful bioremediation, there is an urgent need to take into account various environmental factors, such as the coexistence of pollutants, including heavy metals; a lack of nutrients and non-sterile conditions, among other factors. Particularly, the use of WRF under non-sterile conditions is a big challenge because native bacteria grow faster than fungi, which leads to competition for nutrients and, as a consequence, a decrease in the degradation efficiency of WRF. Therefore, it is imperative to incentivize the application of WRF in culture conditions that are close to the in situ conditions. Some topics have been underexplored and should be a priority for research today, such as the co-remediation of xenobiotic compounds and heavy metals, the symbiotic action of WRF with bacteria and/or plants, the impact of the application of WRF strains and/or their enzymes on ecosystems and their microbial communities and the development of slow-release nutrient promoters used for in situ bioremediation. Additionally, since most contaminated sites are different from each other, it is mandatory to implement a site-specific approach by understanding the fungal mechanisms involved in the degradation of each pollutant under certain environmental conditions. Rigorous research needs to be conducted on the development of simulation models for analyzing the chemical structure of xenobiotics, the environment and the possible products of degradation. A preliminary analysis of all these factors can help us to understand the fate of the fungal biodegradation process.

In the case of ex situ remediation and industrial wastewater treatment, various bioreactor configurations have evolved in the past years. In this regard, whole pellets or immobilized cells/enzymes can be used for the treatment of wastewater containing xenobiotics [174]. However, fungi generally do not grow well in a suspended cell system and are very sensitive to process operations. To overcome such obstacles, immobilization techniques and different traditional carriers for fungal cells and enzymes have been developed [160]. However, the choice of nanomaterials such as carbon nanotubes and carbon nanoparticles for the enzyme immobilization has several advantages over the traditional support, such as minimization of diffusion and maximization of the surface area available for contact between the biocatalysts and the pollutant [180,182]. Future research should focus on the development of nanomaterials for the immobilization of ligninolytic enzymes and also fungal cells with reusability for long-term operation in bioreactors. Furthermore, in the case of the utilization of class II peroxidases for ex situ and in situ remediation is the delivery of H_2_O_2_. Moreover, in order to realize the long-term operation of a bioreactor, it is important to keep stable conditions. Future research should focus on the optimization of parameters such as the selection of appropriate substrates, the reactor design and the determination of the optimal factors for growing WRF. Additionally, the economic feasibility is another obstacle for WRF-based treatments; it indeed is important to develop cost-effective bioreactors.

Finally, in order to successfully apply ligninolytic enzymes such as laccases, LiPs, MnPs, VPLs and DyPs for both ex situ and in situ remediation, further investigation is required, especially to obtain higher levels of these enzymes using DNA recombinant technology.

## 7. Conclusions

Based on the studies reviewed in this work, we corroborate that the unique characteristics of WRF and their enzymes have great potential for the bioremediation of xenobiotics-polluted ecosystems and for the treatment of industrial and hospital wastewater containing toxic xenobiotics such as textile dyes, polycyclic aromatic hydrocarbons and pharmaceuticals, among others. In spite of some obstacles in the way of their practical application, WRF and their enzymes are appropriate, effective and economical tools to implement in bioremediation strategies. Further research is still required to fully explore the possibilities of white rot fungi at the industrial level.

## Figures and Tables

**Figure 1 jof-10-00167-f001:**
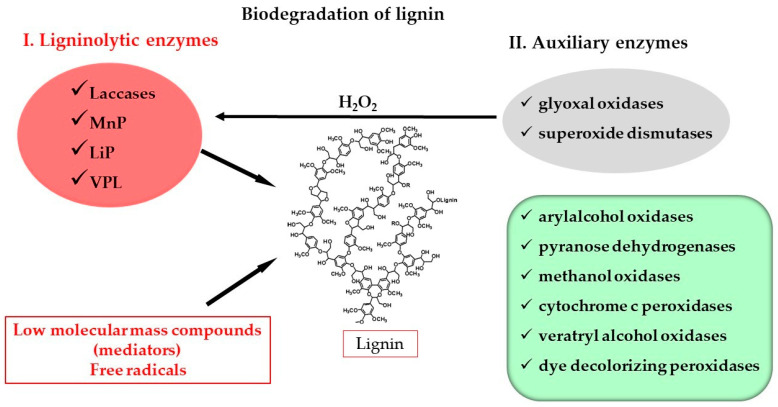
Biodegradation of lignin by enzymes produced by white rot fungi.

**Figure 2 jof-10-00167-f002:**
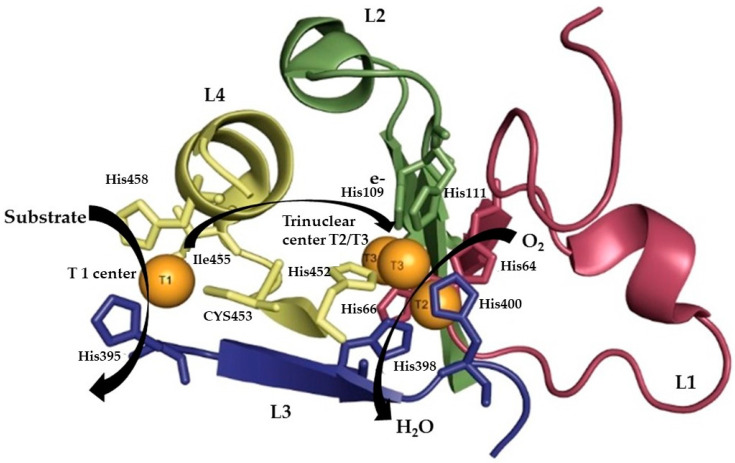
Structure of the active site of laccase from *Trametes versicolor* (PDB entry 1GYC, [76]). The copper binding residues and copper centers are presented in orange. The conserved patterns L1, L2, L3, L4 are shown in red, green, blue, yellow, respectively (modified from Sirim et al. [62]). Black arrows mark the movement of electrons (e^−^), O_2_ and substrates (modified from Giardina et al. [58]).

**Figure 3 jof-10-00167-f003:**
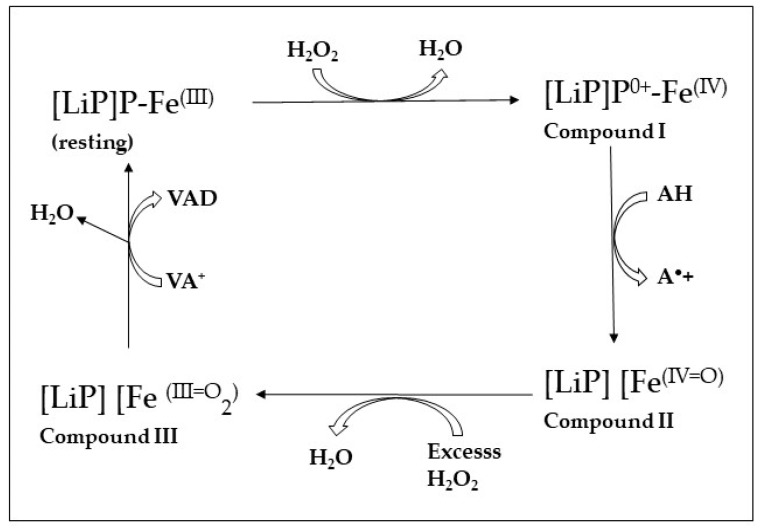
Lignin peroxidase (LiP) mechanism. Porphyrin ring system: P-FE; resting: rest state; nonphenolic aromatic substrate: A; aryl cation radical of A: A•+ (modified from Hodrichter et al. [103]).

**Figure 4 jof-10-00167-f004:**
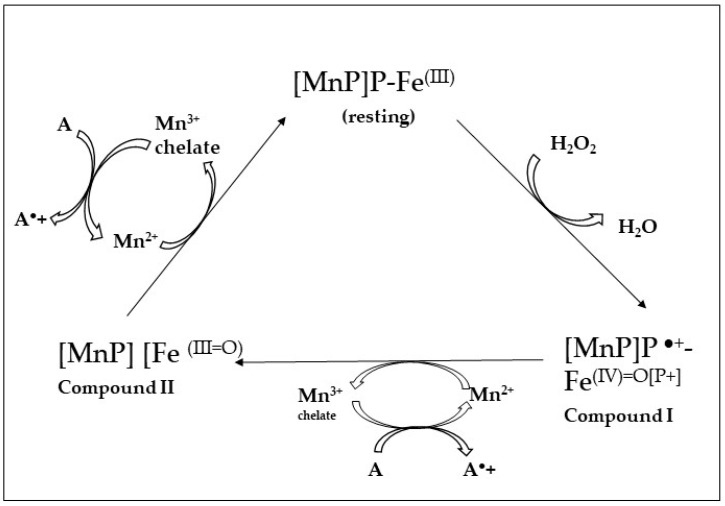
Catalytic cycle of manganese peroxidase (MnP). Porphyrin: P; chelate complex of organic acids: [Mn^3+^ chelate] (Hofrichter et al. [103]).

**Figure 5 jof-10-00167-f005:**
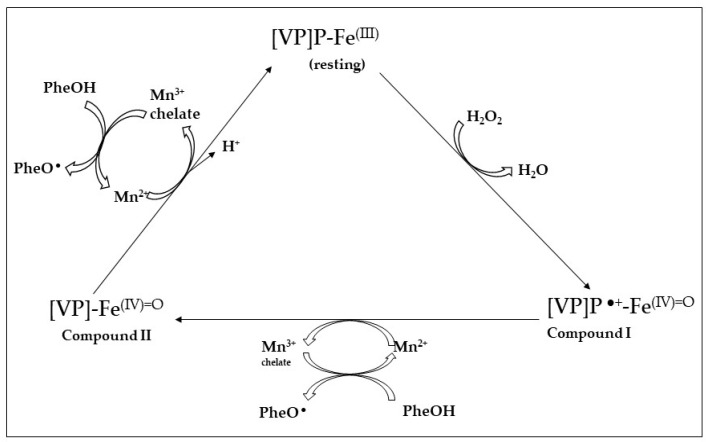
Catalytic cycle of versatile peroxidase (VPLs). Porphyrin: P; non-phenolic high-redox potential aromatic substrate: A, aryl cation radical of A: A•+; phenolic compound: Phe–OH, phenoxyl radical: PheO• (Ruiz-Dueñas et al. [102]).

**Figure 6 jof-10-00167-f006:**
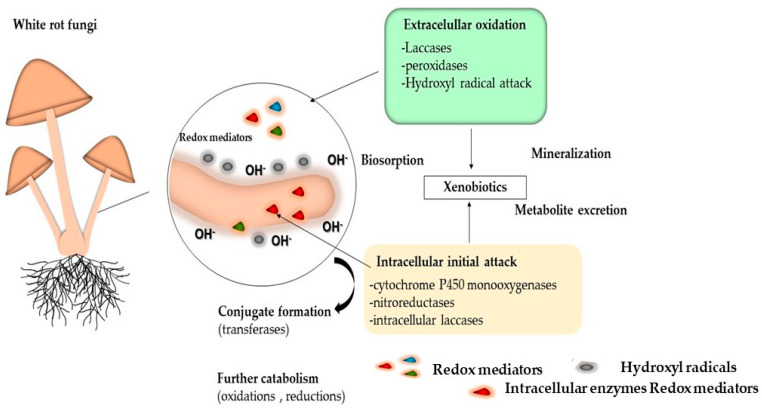
Main mechanism used by fungi to degrade xenobiotics.

## Data Availability

Data are contained within the article.

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
