# Peer review of "White Rot Fungi as Tools for the Bioremediation of Xenobiotics: A Review"

_jof, 2024, doi:10.3390/jof10030167_

Round 1

Reviewer 1 Report

Comments and Suggestions for Authors

White rot fungi as tools are of interest for bioremediation of xenobiotics and other biotechnological applications as well. The authors in the manuscript jof-2783257 performed a review on the white rot fungi and their enzymes such as laccases with activity for xenobiotics degradation.

From the review perspective, there are some shortcomings in the manuscript.

I invite authors to consider adequately describing molecular insights from genome sequence by white rot fungi (Basiodiomycota) to degrade xenobiotics in context of findings on other fungi such as Ascomycota. Compared to ascomycetes, what are the degradation and bioremediation capabilities of xenobiotics by white rot fungi and other basidiomycetous fungi? Has the genome sequence of white rot fungi revealed peculiar xenobiotic-degrading enzymes? Have any distinctive genes involved in mycoremediation been identified by genomic analysis in white rot fungi? Molecular insights from the genome sequence of white rot fungi, if any, should be included in the manuscript.

It would be helpful to improve the presentation of the figures in the manuscript. For instance, in the figure 6 it would be helpful to better clarify the coloured symbols reported in the figure. The authors should consider reformulating the caption.

I hope it helps.

Comments on the Quality of English Language

Minor editing of English language required.

Author Response

Thanks for your suggestions please find my answers in the attached file 

Reviewer 2 Report

Comments and Suggestions for Authors

The present paper is of interest for a large community including microbiologist an biologist as it summarized recent trends for the bioremediation of a large diversity of xenobiotics.

I have just a few concerns as follows:

1. line 102, the sentence is vague : “Their principal characteristics", principal characteristics of what?

2. Lines 109-110 and page 11, lines 456-57, or author should write “white rot fungus” or “white-rot fungus”, they have to choose. The same for brown rot fungus.

3. Line 121 : Rephrase “Due ti its complex structure”

4. Lines 122-123 : reformulate as for instance : “which makes it too large to be adsorbed by the fungus for intracellular attack or degradation…”

5. Lines 134-136 : or some genome do not possess gene encoding laccase for instance.

6. Please revise the entire section Lines 142- 148 : as glyoxal oxidases appears two time with different EC number and with typing error for the second (glyoxal oxidases), I am not sure that the second glyoxal oxidase are intracellular, please, check it. Correct spelling of aryl alcohol oxidases. You can provide their CAZy number as you did for haeme-peroxidases and cite the CAZy reference (Levasseur A, Drula E, Lombard V, Coutinho PM, Henrissat B. Expansion of the enzymatic repertoire of the CAZy database to integrate auxiliary redox enzymes. Biotech. for Biofuels 6:41 [PMID : 23514094]=.

7. ad a s for laccase in the title of section 3.1 line 170. AA1 for laccases.

8. Please, illustrate names of natural natural compounds line 196. And please give the full name and abbreviation after for ABTS. Line 197. Idem for “trinuclear copper cluster” : TNC lien 207.

9. Section lines 239-248, please add a “;” or a “,” between the strain, but choose one.

10. Please add the reference of CAZy after ref 95 line 251.  Line 257, ad a value to the average redox potential of class II peroxidase.

10. Important remark : all along the text, when you write a fungal name for the first time, it should appear as the whole and after abbreviated,  like Trametes versicolor and then T. versicolor. There is no rule in your text and appears randomly for all the fungal name. This is boring for a microbiologist.

11. Cancel “The”  at “the VA” line 288.

12 : Line 296 “long range electron transfer” should appear as LRET as it was already define earlier in the text.

13. Line 317 : “basidiomycetes”. And line 327, is missing a word in the sentence after “P. chrysosporium…”

14. Line 415 : please define Co (carbon monoxide”. And later : DDT, BPA, PC¨, TNT…

15 No legend and title was find for Table 1. Check Table 1 text : Oxidation of phenolic (capital letter), etc… -“Detection of quinolones” put at the right place.

16: Lines 547-549: Authors talk about fungal immobilization and for enzymes?

17 : Lines 553-55 : give at least one reason why aquatic fungal fungus are interesting for their enzymes?

18. Lines 571-586 : authors do not talk about the problem to provide and deliver hydrogen peroxide to the class II peroxidases?

19: Section 5.1 should be more concise as it is a list of decolorization by several fungi without giving the plus of each system. Are they only decolorized, what about the residual toxicity or the molecules obtained?

20. 727 : “solid state fermentation” (no capital), the same “perfluorooctane” line 865.

21 : check Table 2 : name of the fungus as usual; abbreviation of in full…

22. : Lines 913-914 : this is the title of a new section? Please check it.

23. : The delivery of hydrogen peroxide is also a challenge , line 988-990.

Author Response

Thanks for your suggestions. Please find my comments in the attached file. 

Round 2

Reviewer 2 Report

Comments and Suggestions for Authors

The authors have significantly revised the paper according to my comments. I thanks then for their corrections and confirm that the paper is acceptable for publication.